# Advanced Imaging for Robotic Bronchoscopy: A Review

**DOI:** 10.3390/diagnostics13050990

**Published:** 2023-03-05

**Authors:** Nakul Ravikumar, Elliot Ho, Ajay Wagh, Septimiu Murgu

**Affiliations:** 1Interventional Pulmonology, Section of Pulmonary and Critical Care, Department of Medicine, University of Chicago, Chicago, IL 60637, USA; 2Interventional Pulmonology, Division of Pulmonary & Critical Care Medicine, Department of Medicine, Loma Linda University, Loma Linda, CA 92354, USA

**Keywords:** robotic bronchoscopy, cone-beam CT, C-arm based tomosynthesis, CIOS spin, O-arm CT, ventilator strategies, navigational bronchoscopy, CT-to-body divergence

## Abstract

Recent advances in navigational platforms have led bronchoscopists to make major strides in diagnostic interventions for pulmonary parenchymal lesions. Over the last decade, multiple platforms including electromagnetic navigation and robotic bronchoscopy have allowed bronchoscopists to safely navigate farther into the lung parenchyma with increased stability and accuracy. Limitations persist, even with these newer technologies, in achieving a similar or higher diagnostic yield when compared to the transthoracic computed tomography (CT) guided needle approach. One of the major limitations to this effect is due to CT-to-body divergence. Real-time feedback that better defines the tool–lesion relationship is vital and can be obtained with additional imaging using radial endobronchial ultrasound, C-arm based tomosynthesis, cone-beam CT (fixed or mobile), and O-arm CT. Herein, we describe the role of this adjunct imaging with robotic bronchoscopy for diagnostic purposes, describe potential strategies to counteract the CT-to-body divergence phenomenon, and address the potential role of advanced imaging for lung tumor ablation.

## 1. Introduction

Lung nodules are discovered on chest computed tomography (CT) images over 25% of the time, and in approximately 1.6 million people annually in the US [1]. Incidental and screening-detected nodules are expected to rise with the increased use of chest CT for the diagnosis of advanced lung diseases including emphysema, bronchiectasis, interstitial lung disease, as well as the liberalized guidelines for lung cancer screening [2]. Approximately 95% of all nodules detected on imaging are benign, but in high-risk populations (i.e., age 50–80 years, >20 pack year smoking, smoking within the last 15 years, personal history of cancer, family history of lung cancer) a timely and minimally invasive diagnostic modality would expedite the management of a potentially malignant nodule while minimizing testing for patients with a benign nodule. The approach utilizing bronchoscopy as a diagnostic method is the preferred technique for evaluating lung nodules suspicious for lung cancer due to the safety profile and because it allows for concurrent staging via endobronchial ultrasound-guided transbronchial needle aspiration (EBUS-TBNA) of the mediastinal, hilar, and interlobar lymph nodes [3].

Recent advances in robotic-assisted bronchoscopy (RAB) have enabled chest physicians to safely navigate within the lung and sample parenchymal pulmonary lesions (PPL) with more confidence and increasing accuracy. Advantages of these navigation platforms are enhanced maneuverability, farther reach, and increased stability of the bronchoscope as compared to conventional guided bronchoscopy, but their diagnostic yield remains to be improved [4,5,6,7,8]. These platforms guide the bronchoscopist by creating a virtual pathway to a target lesion, but limitations in the accuracy remain due to the absence of real-time guidance. Additional technologies such as radial EBUS (rEBUS), augmented fluoroscopy with C-arm based tomography (CABT), or cone-beam CT (CBCT) may be used for real-time confirmation of lesion location and confirmation of “tool in lesion” prior to sampling.

### CT-to-Body Divergence

Regardless of the navigation platform used, a planning CT is created with a thin-cut chest CT scan which is obtained with the patient at the end of an inspiratory hold maneuver with lung volume close to total lung capacity (TLC). This maneuver helps enhance the airway detection (segmentation) on the planning software and offers a better navigation path selection. The target PPL is identified on the virtual airway tree and its relationship to the airway is defined. During the procedure, the virtual reconstruction of the airway and the relative location of the target PPL is synchronized with the position of the robotic bronchoscope and the patient’s anatomy in vivo via a process called registration (Figure 1). Once registration is complete, the operator navigates to the target lesion by using the virtually mapped pathway. However, the lung volume of the CT scan used for procedural planning (at full inspiration, breath hold, and nearing total lung capacity) is different than the patient’s lung volume at the time of the procedure (mechanical breathing under general anesthesia and paralysis with volume nearing functional residual capacity), potentially leading to a phenomenon known as CT-to-body divergence (CTBD). CTBD is the difference between PPL location at the time of planning CT and at the time of procedure. Other than the differences in lung volume, the development of atelectasis during the procedure related to ventilation strategies, oxygen supplementation, and manipulation of the airway with tools, can lead to further worsening of CTBD [9].

Hence, secondary to this phenomenon, the navigation system’s feedback could be misleading and provide a false reassurance of successful localization to a virtual target when in fact the scope may be off the real target (Figure 2). This is particularly true for nodules in the lower lobes due to more prevalent atelectasis during general anesthesia.

CTBD has been evaluated in multiple studies. Chen and colleagues performed two chest CT scans for patients undergoing navigation bronchoscopy at full inspiration and at tidal volume breathing, which demonstrated a mean deviation of 17 mm of nodules between two scans [10]. A study using the Ion^TM^ RAB platform identified that CTBD was present in 50% of the nodules, with divergence defined as an overlap of less than 10% between the target location on the preoperative CT and the target location during real-time mobile 3D imaging; this rate increased to 60% when CTBD was redefined based on 10 mm distance between virtual and real-time targets [11]. Similarly, an average divergence of 15 mm was corrected after tomosynthesis in a study that used Illumisite^TM^ (Medtronic, Minneapolis, MN, USA) [12].

For this purpose, there is an increased effort in the development of advanced imaging to support RAB. The goal is to provide real-time image guidance during RAB to better define the lesion and its proximity to the bronchoscope. The combined approach with advanced imaging techniques and RAB for PPL sampling may improve the diagnostic yield. In the next section, we review the evidence of using advanced imaging for PPL sampling during advanced bronchoscopy, with emphasis on robotic technologies.

## 2. Augmented Fluoroscopy

Augmented fluoroscopy is used to define the relationship between the location of the nodule in real-time and a preprocedural CT scan. This is evaluated by performing tomosynthesis intra-operatively to provide a “local registration” of the nodule. Tomosynthesis is performed using a conventional C-arm fluoroscopy that obtains images in multiple planes and then is matched to the preoperative CT images. This attempts to correct for CTBD by updating the position of the target lesion and improving real-time localization of the robotic bronchoscope with fluoroscopic imaging in three dimensions [13,14,15,16].

This imaging modality has been used with electromagnetic navigation bronchoscopy (ENB) in the past with favorable results. Fluoroscopic ENB (Illumisite^TM^, Medtronic, Minneapolis, MN, USA) is one such technology that uses tomosynthesis via a conventional fluoroscopy arm to visualize and re-register the target lesion on real-time imaging (Figure 3). Aboudara and colleagues conducted a retrospective review of all procedures that used the superDimension iLogic 7.2 ENB platform (superDimension, Medtronic) combined with fluoroscopic tomosynthesis (F-ENB). Diagnostic yield was compared between F-ENB and standard ENB (S-ENB). The primary outcome of diagnostic yield was significantly higher in the F-ENB group at 79% compared to 54% in the S-ENB group (*p* < 0.05). The median divergence was 12 mm in this study [15]. A single-center, prospective, observation study looked at outcome data for 100 patients who had biopsies performed by this navigation platform with continuous real-time guidance (Illumisite™; Medtronic, Minneapolis, MN, USA) and reported a diagnostic yield of 83% with sensitivity for malignancy at 71% [17]. To date, there are no published studies of integrating this technology with any of the available RAB platforms.

### 2.1. LungVision™

LungVision™ (Body Vision Medical Ltd., Ramat Ha Sharon, Israel) uses augmented fluoroscopy and artificial intelligence (AI) to assist with intra-operative nodule localization and generates a navigational pathway by combining the preprocedural CT scan images with real-time fluoroscopy images. This AI-based technology has an additional capability to track the location of the scope and different tools in real-time to adjust with the divergence [14]. A single-center study using this technology with conventional bronchoscopy demonstrated lesion localization success of 96.1%. The average distance between lesion locations was shown by LungVision™ augmented fluoroscopy and the actual location measured by CBCT was 5.9 mm (range: 2.1 to 10.0 mm). Diagnostic yield at the index procedure was 78.4%. Diagnostic accuracy assessed at 12 months follow-up was 88.2%. Average CTBD was 14.5 mm [18]. A multicenter study of 55 patients demonstrated a nodule localization success rate at 93% with an overall diagnostic yield of 75.4% based on an immediate rapid on-site pathology report [13]. A recent retrospective study involving 45 patients undergoing navigation bronchoscopy for pulmonary nodule using the Monarch™ robotic platform, rEBUS, and the Body Vision system yielded an immediate diagnostic yield of 84% and a final diagnostic yield of 91% (Figure 4) [19].

### 2.2. CIOS Spin

The CIOS 3D Spin Mobile (Siemens Healthineers) is a compact C-arm that is electronically rotated around the patient’s chest by 100 degrees to generate a 3D CT image or 2D fluoroscopy (Figure 5). A prospective, single-center, single-arm pilot study involving 30 patients to evaluate the clinical utility and performance of the CIOS 3D Mobile Spin system in conjunction with the Ion Endoluminal System showed 100% ability to navigate to the lesion, a diagnostic yield of 93%, and an overall sensitivity for malignancy of 91% [11]. In a smaller study, which evaluated 10 lesions in eight patients using the Ion^TM^ Platform in conjunction with the CIOS Mobile 3D spin, tool-in-lesion was confirmed in 90% of patients. The relationship between the biopsy tool and lesion was improved in three instances after the subsequent redeployment of the tool, based on feedback from the intra-operative portable CT imaging [20].

## 3. Cone-Beam CT

Cone-beam CT (CBCT) is a technology that uses a compact CT system with a ceiling- or floor-mounted C-arm that can be utilized during bronchoscopy to provide real-time feedback of the bronchoscope or tool location. CBCT uses a flat panel detector system made from cesium iodide scintillators as a detector for cone-shaped (wide collimation) X-ray beam from the X-ray source. Three-dimensional images are then produced with a reconstruction algorithm. The imaging is reviewed during the procedure to assess bronchoscope, tool, and target lesion locations and help the operator determine if adjustments are needed to reach the target lesion (Figure 6). CBCT differs from conventional multi-detector CT with respect to a lower radiation dose and time of image acquisition (typical scanning time of 5–10 s). However, a significant limitation of this technology is the current cost, which may make it difficult to obtain for most pulmonologists [22].

### 3.1. CBCT with Conventional and Non-RAB Navigation Platforms

CBCT has been previously used in non-robotic electromagnetic platforms with improved success in localization and diagnostic yield. Pritchett and colleagues combined CBCT with the SuperDimension system to achieve lesion localization and correction. The authors found that with additional imaging, three-dimensional target overlap improved from 59% to 83% after location correction. In the same study, the percent of cases without target overlap decreased from 31% to 5% after location correction [23]. Although this study did not report diagnostic yield, another retrospective review of 75 patients who underwent ENB combined with CBCT found that the overall diagnostic yield was 83% [16]. A study comparing ENB alone and ENB-CBCT showed an improved diagnostic yield to 74% from 51% (*p* = 0.05) [24]. In another study of 20 patients that combined CBCT with thin/ultra-thin bronchoscopy, there was a 25% absolute increase in the diagnostic yield of PPL sampling [25]. Similarly, CBCT use with conventional bronchoscopy and ultra-thin bronchoscopy (UTB) for PPL sampling has shown significantly improved diagnostic accuracy with the combination of the CBCT + UTB + rEBUS group compared with the rEBUS group alone. The diagnostic yield with CBCT + UTB + rEBUS vs. the conventional bronchoscopy + rEBUS group was 85% and 44%, respectively, a difference that was statistically significant. The diagnostic yield of CBCT with conventional bronchoscopy was 68% [26].

### 3.2. CBCT with RAB Navigational Platforms

As of this writing, there are two FDA-approved robotic platforms available in the US. The Monarch^TM^ platform by Auris Health (now Johnson and Johnson) and the Ion^TM^ platform by Intuitive Surgical. The Monarch^TM^ platform uses electromagnetic navigation with real-time vision input from a camera situated at the tip of the bronchoscope (optical pattern recognition), while the Ion^TM^ platform uses a fiber optic shape-sensing technology to guide navigation. The Monarch^TM^ platform has the advantage of constant peripheral vision that helps guide tool articulation during the procedure and monitor for any complications such as bleeding, while the Ion^TM^ platform has an advantage of a smaller bronchoscope (3.5 mm outer diameter vs. the Monarch 4.4 mm outer diameter) that may allow for more peripheral access. However, these two technologies have not yet been compared in a head-to-head trial to see if these differences have any impact on the diagnostic yield.

#### 3.2.1. Ion™ Robotic Platform Combined with CBCT

Two studies reported a diagnostic yield of 81% with use of the Ion platform alone with a sensitivity of 79% and 87% for malignancy, respectively [27,28]. Recently, there has been an increased combined use of CBCT with the Ion platform. A prospective study of 52 consecutive patients who underwent robotic bronchoscopy with the Ion™ platform combined with CBCT reported sensitivity of 84% for malignancy and an overall diagnostic yield of 86% [29]. Another recent study that combined the Ion system with CBCT and rEBUS showed a diagnostic accuracy of 91% with a sensitivity of 88% for malignancy in 198 patients. REBUS did not confirm the lesion in 12% of the cases that were biopsied with CBCT guidance [30].

#### 3.2.2. Monarch™ Robotic Platform Combined with CBCT

Using strict definitions, studies reported a diagnostic yield of 74%–77% with the use of the Monarch robotic platform alone with a sensitivity for malignancy of 82% in the BENEFIT Trial [4,5]. A retrospective analysis of 20 patients who underwent robotic bronchoscopy with the Monarch™ platform and CBCT showed 100% navigational success demonstrating the tool in lesion and sensitivity of 86% for the diagnosis of malignancy [31]. The use of RAB has now expanded to include dye marking of lung lesions for subsequent lung resection. RAB’s ability to inject dye closer to the target lesion could aid in intra-operative visualization of dye, better margin determination, and more lung preservation [32]. With these additional indications for RAB, even more precise target localization is indicated and CBCT can assist with this RAB platform to improve its precision.

## 4. O-Arm CT (OACT)

O-arm CT is a mobile CT device with an O-shaped gantry that encircles the patient and can be moved as a unit towards the head or foot of the bed. The added benefits of this device are its mobility, smaller footprint, and the shape of the gantry (can be opened halfway to a shape of “C”) that helps with the positioning of the patient during the procedure. The O-arm obtains and reconstructs images in multiple planes. This device has been previously used for the localization of non-palpable nodules during video-assisted thoracoscopic surgery [33]. Recently, O-arm CT has been described as a technically feasible option for use during ENB in a study with six patients [34]. A recent retrospective study by Chambers et al. yielded a diagnosis in 77% of patients. Tool-in-lesion was again confirmed by O-arm CT in a high number of cases at 97%, but diagnostic yield was not improved. The authors reported seven additional cases that did not have enough cells for cell block preparation but were suspicious for malignancy and eventually diagnosed with malignancy at surgical resection. Counting these cases, diagnostic yield would have been 86% [35].

These studies suggest that the use of additional imaging with CBCT, augmented fluoroscopy, and OACT across a variety of navigational platforms is helpful in obtaining real-time feedback that aids navigation in close to 100% of the lesions, with possible improvements in the diagnostic yield and accuracy. See Table 1 for a description of the studies using real-time imaging at the time of navigational bronchoscopy with the above-mentioned platforms and their reported yield and adverse events.

## 5. Bronchoscopic Tools to Improve Diagnostic Yield

Bronchoscopic tool selection may influence diagnostic yield. While several advanced imaging strategies have been employed to ensure that a bronchoscopic tool is in a target lesion, this does not necessarily ensure that a diagnostic sample or adequate sample has been obtained. A post-hoc analysis of the NAVIGATE trial performed by Gildea and colleagues determined that the use of extensive biopsy tool strategies including an aspirating needle may provide a higher true positive rate of diagnosis without increasing complications. The authors found that true positive rates were highest when using the aspirating needle (86.6%) and biopsy forceps (86.9%) [36]. Notably, all the robotic bronchoscopy studies published to date have used needle aspiration in 100% of cases. Therefore, the use of needle aspiration in these studies may have contributed to a higher diagnostic yield seen with these technologies when compared with prior studies of advanced bronchoscopy techniques.

## 6. Strategies to Reduce CTBD

Several investigators have addressed the pervasive issue of CTBD and methods to mitigate it. Ventilator strategies have been employed to minimize this effect. A routine CT chest is ideally performed at total lung capacity (TLC); however, this cannot be consistently reproduced intra-operatively, despite the use of positive pressure ventilation and recruitment maneuvers. Air follows the path of least resistance, resulting in better ventilation of non-dependent lung regions. In addition, diaphragmatic movement, airway distortion, segmental occlusion with the bronchoscope, airway bleeding, and time-dependent atelectasis result in the worsening of CTBD and variable nodule location. At least two studies reported that CTBD was worse in the lower than in the upper lobes, with values in the lower lobes >20 mm [10,11].

Several studies evaluated the effects of ventilation strategies on atelectasis and tool-in-lesion confirmation using CBCT. In one study of 50 subjects with 27 nodules in the conventional group and 25 nodules in the lung navigation ventilation protocol (LNVP) group, the authors found that the LNVP demonstrated markedly reduced dependent and sublobar/lobar atelectasis compared with conventional ventilation [37]. The LNVP included rapid intubation using an 8.5 endotracheal (ET) tube or larger along with paralysis using a non-depolarizing muscle relaxant and applying the lowest tolerable fraction of inspired oxygen (FiO_2_). Additionally, recruitment maneuvers were utilized with a tidal volume of 10–12 cc/kg ideal body weight. A differential PEEP was applied for upper/middle lobe lesions (10–15 cmH_2_O) and lower lobe lesions (15–20 cmH_2_O). This study noted that dependent and lobar atelectasis was higher in the conventional group compared to the LNVP group (*p* < 0.05), and there was also a trend toward improved diagnostic yield in the LNVP group (92% vs. 70%, *p* = 0.08). Pritchett and colleagues similarly described a ventilation protocol strategy based on anesthesia literature to minimize atelectasis intra-procedurally [38]. The algorithm proposed by the authors can be reviewed in Table 2. Another randomized study involving 76 patients also evaluated ventilator strategies to prevent atelectasis under general anesthesia. The control group was ventilated via the laryngeal mask airway (LMA) with FiO_2_ of 1.0 and 0 cmH_2_0 PEEP versus ventilation via ET tube with FiO_2_ titrated as low as possible for oxygen saturation of more than 94% and PEEP of 8–10 cmH_2_O in the intervention group. Atelectasis was then studied on CT and radial EBUS imaging obtained at two different time intervals. This study showed a reduction in any atelectasis formation from 84% to 29% and bilateral atelectasis from 71% to 8%. There was no difference in the complication rates between the groups, and no cases of pneumothorax or pneumomediastinum occurred. This study, however, did not evaluate for the difference in diagnostic yield between the two strategies [39]. These strategies suggest that using a higher PEEP, lower FiO_2_, and tidal volumes in the range of 6–8 cc/kg ideal body weight can help prevent atelectasis and thereby counteract CTBD. In addition, a recent study combining the use of CIOS spin with the Ion robotic platform evaluated atelectasis prevention by maintaining the patient in a lateral decubitus position (target lesion side up). This study demonstrated no atelectasis development in all patients. However, the study did not report on whether this led to an increase in diagnostic yield, and further studies are required to evaluate this approach [40]. This is relevant as there is increasing concern for misinterpretation of R-EBUS images due to atelectasis. Experienced operators, however, could distinguish the lesions from atelectasis based on several features including margins, size, absence of blood vessel, absence of linear-discrete air bronchograms, or heterogeneous sonographic pattern.

## 7. Therapeutic Potentials with Advanced Bronchoscopy and Augmented Imaging

### 7.1. Lung Tumor Ablation

Due to the detection of early-stage lung cancers, it is expected that we will see a rise in the detection of inoperable early-stage lung cancers in patients with multiple co-morbidities or poor performance status [41]. The current standard of care for early-stage lung cancer is surgical resection. However, in patients with a resectable disease that is medically inoperable, alternative treatments exist, such as stereotactic body radiation therapy (SBRT) [42,43,44,45,46,47,48]. SBRT has its limitations in terms of toxicity including posttreatment fibrosis, the limitation of re-treatment in patients with previous radiation therapy, and proximity of the lesion to vital anatomical structures. As another option, multiple societies including CHEST, the NCCN (National Comprehensive Cancer Network), and the Society of Thoracic Surgeons suggest image-guided thermal ablative (IGTA) therapy for inoperable early-stage lung cancer [45,49,50,51]. IGTA is a form of local ablative therapy that includes radiofrequency ablation (RFA), microwave ablation (MWA), and cryotherapy ablation (CA) [52,53,54]. Previously, this form of treatment has been performed with a percutaneous approach using CBCT, ultrasound (US), and CT fluoroscopy; however, significant adverse events have been reported including pneumothorax in up to 45%, chest tube insertion required in about 20% of patients, and pulmonary hemorrhage has been reported in 6% of the cases with 2% requiring intervention [55].

Considering the available data on treatment efficacy and high rate of complications from a percutaneous approach, bronchoscopy-guided local ablative therapy is a potential option. Several preclinical studies are available that have used these technologies via bronchoscopy to assess safety [56,57,58,59,60]. Several human trials have studied this modality via the bronchoscopic approach [61,62,63,64,65,66]. A CT imaging- bronchoscopy guided-cooled RFA performed in 20 patients with 28 lesions showed a local control rate in 82% of lesions with a progression-free survival of 35 months [61]. A study using navigational bronchoscopy along with R-EBUS in 13 patients with 14 tumors showed a complete ablation rate of 78%, with a median progression-free survival of 33 months and pneumothorax noted in two patients [62]. Another study combining ENB with CBCT for the transbronchial microwave ablation of 30 lesions in 25 patients showed 100% technical success. There was no evidence of disease progression in all 30 nodules at the 12-month mark, and the rate of pneumothorax requiring chest tube placement was 6% [63]. One study combining ENB with CBCT reporting on the safety and feasibility reported a total of two deaths with one possible procedure-related death and no other events of pneumothorax or hemoptysis [65]. In another such study combining ENB and CBCT in patients with multiple pulmonary nodules, 96 lesions were treated with MWA with a 3% pneumothorax rate [66].

### 7.2. Perspective on the Future of Bronchoscopy-Guided Lung Tumor Ablation

The above-mentioned studies suggest that a bronchoscopy-guided approach for ablation of peripheral lung tumors is technically feasible with the potential added benefit of lower complications. The availability of additional imaging to provide real-time feedback to confirm the tool in lesion might help increase the technical accuracy of an ablative procedure that could be comparable to the percutaneous transthoracic approach, but with lower complication rates. However, further studies comparing the safety of therapy and long-term outcomes are required. Currently, multiple studies are ongoing to evaluate this modality including NCT03490890 and NCT05053802 [67,68]. NCT05299606, a prospective, multicenter trial combining robotic bronchoscopy with microwave ablation, is currently recruiting to evaluate navigation and ablation success [69]. NCT05281237, a prospective study, is being listed to study the effect of MWA of lung tumors with CBCT-guided navigation bronchoscopy [70].

## 8. Conclusions

The current data suggest that the multimodality approach of using RAB in combination with advanced imaging leads to an improvement in lesion localization and “tool in lesion” confirmation when sampling PPL. Prospective, controlled studies are needed to evaluate the impact of this combined approach on diagnostic yield. As the technology for advanced imaging for RAB continues to improve and instruments that allow for real-time imaging during lung lesion sampling develop, localization success and diagnostic accuracy for PPL sampling will hopefully continue to increase. The continued development of advanced imaging to support RAB is pivotal as the assurance of accurate localization will be crucial when pursuing locally ablative and therapeutic techniques for future treatments of PPL.

## Figures and Tables

**Figure 1 diagnostics-13-00990-f001:**
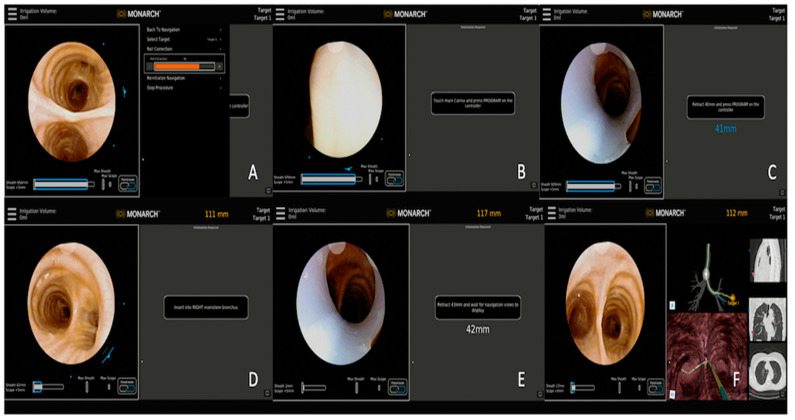
Steps involving registration process. (**A**) Roll correction of the bronchoscope, (**B**) touch main carina to begin registration, (**C**) retract scope back to 40 mm, (**D**) navigate to the contralateral mainstem bronchus (right side in this case), (**E**) retract scope back to the distance as recommended by the navigation device, and (**F**) begin navigation to the left upper lobe nodule.

**Figure 2 diagnostics-13-00990-f002:**
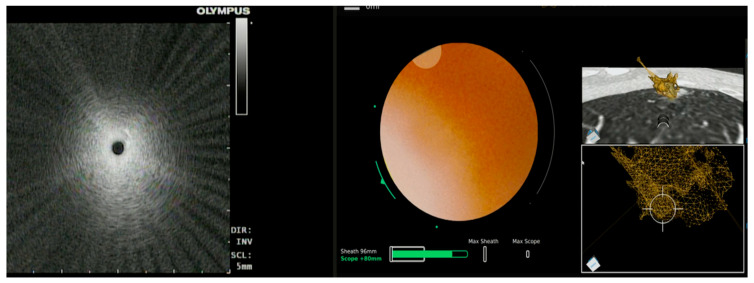
Navigation software feedback showing alignment of the bronchoscope in line with the nodule, but no ultrasound image obtained with radial EBUS.

**Figure 3 diagnostics-13-00990-f003:**
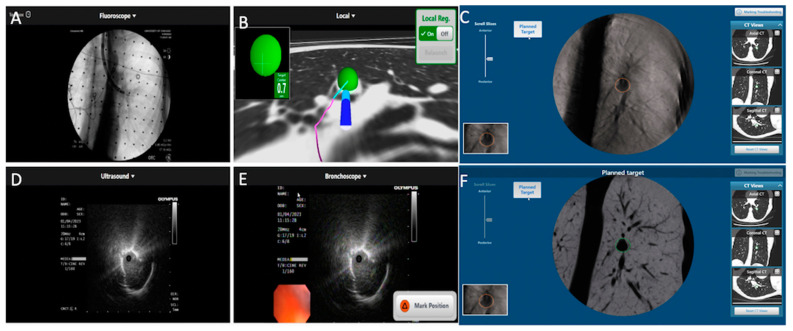
Illumisite^®^ (Medtronic) fluoroscopic navigation during bronchoscopy. (**A**) Fluoroscopy image to guide sampling, (**B**) bronchoscope in alignment with the lesion after tomosynthesis and updated lesion location, (**C**,**F**) planned target location visualized after tomosynthesis spin, and (**D**,**E**) radial EBUS imaging.

**Figure 4 diagnostics-13-00990-f004:**
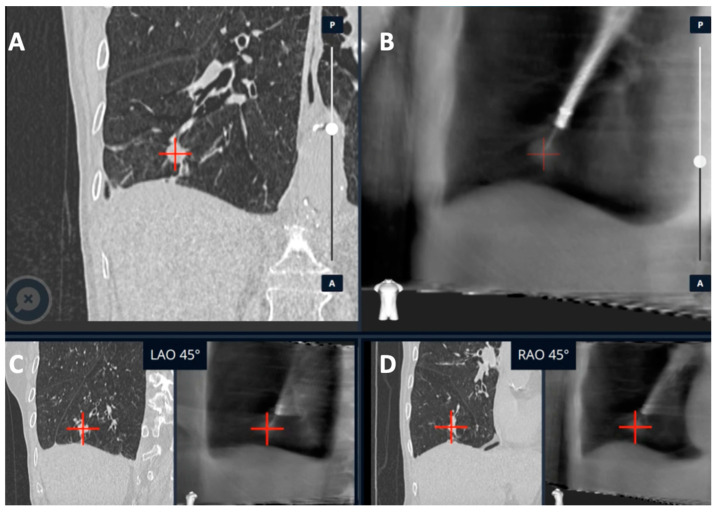
LungVision’s C-arm-based tomosynthesis (CABT) imaging. (**A**) Preoperative CT with lesion. (**B**–**D**) Tool in lesion confirmation obtained in multiple planes.

**Figure 5 diagnostics-13-00990-f005:**
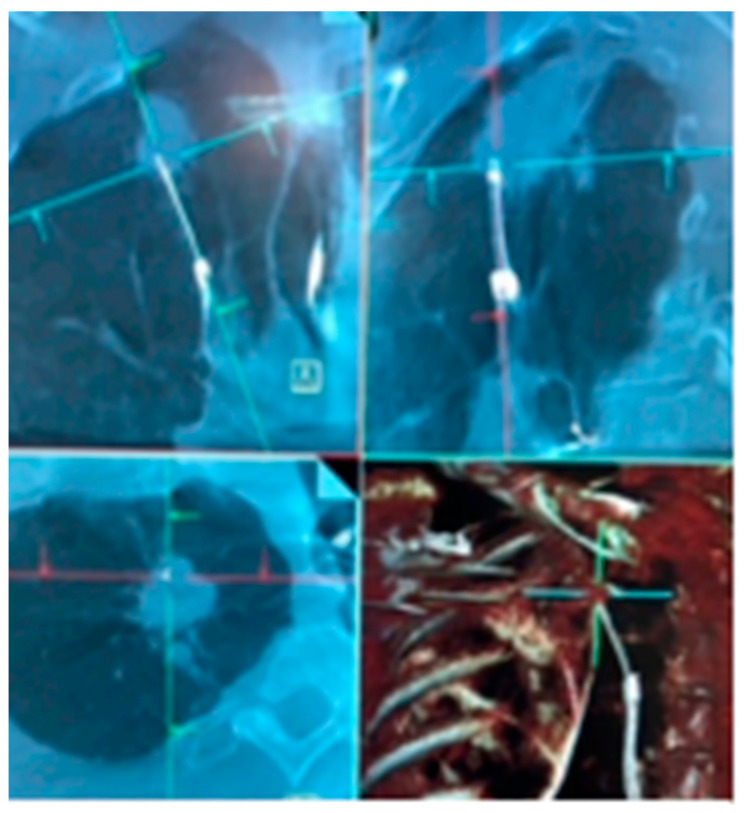
Cios Spin^®^ (Siemens Medical Solutions) mobile 3D C-arm imaging during Flexible Bronchoscopy (Olympus BF-P190) demonstrating tool in lesion confirmation prior to sampling. Reprinted with permission from AME publishing company [21].

**Figure 6 diagnostics-13-00990-f006:**
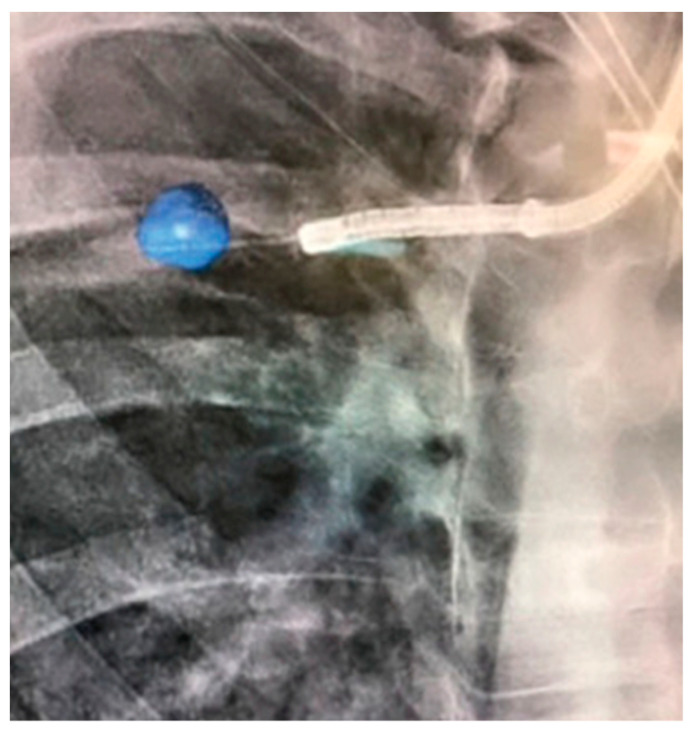
CBCT (Philips Healthcare) imaging during Monarch^®^ Robotic Bronchoscopy with real-time location of the lesion in blue. Reprinted with permission from AME publishing company [21].

**Table 1 diagnostics-13-00990-t001:** Description of studies utilizing additional imaging with robotic bronchoscopy and electromagnetic navigation bronchoscopy.

Publication	Device	Description	Sample Size	Diagnostic Yield	Adverse Events
Aboudara et al. [15]	SuperDimension, Medtronic	Comparison of Standard ENB to Fluoroscopic ENB.	90 lesions (S-ENB) vs. 59 lesions (F-ENB)	79% (F-ENB) vs. 54% (S-ENB). *p* < 0.005. Mean divergence of 12 mm	Pneumothorax (1.9% vs 1.5%)
Avasarala et al. [17]	Illumisite™; Medtronic	Real-time guidance with digital tomosynthesis corrected navigation system	100 lesions	83% overall yield with 71% sensitivity for malignancy (52/73)	Pneumothorax—3%. Bleeding requiring intervention—2%
Cicenia et al. [13]	LungVision; Body Vision Medical LTD.	Real-time fluoroscopic images with integration of images from preop CT.	55 patients	93% nodule localization success; DYi: 75% yield on ROSE	No adverse events
Pritchett [18]	LungVision; Body Vision Medical LTD.	Real-time fluoroscopic guidance for navigation and biopsy with intra-op co-relation using CBCT	51 patients	Localization success: 96%, DYi 78%, and DA: 88%. Average divergence of 14.5 mm	No adverse events
Hedstrom et al. [19]	Monarch™ robotic platform with lung vision	Robotic platform for navigation with CABT from Lung vision for intra-procedural real-time guidance	45 patients	DYi: 84% DA: 91%	Pneumothorax: 8% (4/45)
Kalchiem-Dekel et al. [20]	Ion™ robotic platform with CIOS	Robotic platform for navigation with 3D multiplanar fluoroscopy for intra-procedural real-time guidance	10 lesions	Tool in lesion: 90%. Tool correction in 30% lesions with real-time imaging. DY not reported	-
Reisenauer et al. [11]	Ion™ robotic platform with CIOS	Robotic platform for navigation with 3D multiplanar fluoroscopy for intra-procedural real-time guidance	30 lesions	DYi: 93%. Average divergence: 10 mm in upper lobe 20 mm in lower lobe	No adverse events
Pritchett et al. [16]	SuperDimension, Medtronic with CBCT	ENB system for navigation with intra-procedural CBCT. No rEBUS for any cases	93 lesions	DY: 83% DA: 93%	Pneumothorax: 4%
Kheir et al. [24]	SuperDimension, Medtronic with CBCT	Standard ENB vs. ENB-CBCT	31 patients (ENB) vs. 31 patients (ENB-CBCT)	DY: 74% (ENB-CBCT) vs. 51% (ENB)	Total adverse events (6.5%)—no difference between groups
Benn et al. [29]	Ion™ robotic platform with CBCT	Robotic platform for navigation with intra-operative CBCT for biopsy tool guidance	59 lesions	DY: 83% DA: 86%	Pneumothorax: 3.8%
Styrvoky et al. [30]	Ion™ robotic platform with CBCT	Robotic platform for navigation with intra-operative CBCT for biopsy tool guidance	209 lesions	DA: 91%	Pneumothorax: 1%
Cumbo-Nacheli et al. [31]	Monarch™ robotic platform with CBCT	Robotic platform for navigation with intra-operative CBCT for biopsy tool guidance	20 lesions	Sensitivity for malignancy: 86%	-

Abbreviation: 3D: 3-dimensionsal, CABT: C-arm-based tomosynthesis, CBCT: cone-beam CT, DA: diagnostic accuracy at 12 months, DY: diagnostic yield, DYi: diagnostic yield at index procedure, ENB: electromagnetic navigation bronchoscopy, ROSE: rapid onsite pathology, rEBUS: radial endobronchial ultrasound.

**Table 2 diagnostics-13-00990-t002:** Anesthesia recommendations for advanced guided bronchoscopy. Adapted from Pritchett et al. [38].

Procedural Step	Recommendations
Pre-Oxygenation	Avoid FiO_2_ of 1.0 and use a lower FiO_2_ of 0.6–0.8.
Anesthesia Type	TIVA with Propofol and paralytics.
Intubation	Use larger endotracheal tube if able (≥8.5 mm). Use non-depolarizing muscle relaxant.
Post-Intubation	Perform 4 recruitment maneuvers as able. Maintain FiO_2_ at lowest tolerated level for saturations of above 90% with PEEP up to 10–12 cm H_2_O. Use a tidal volume of 8–10 cc/Kg of ideal body weight.
Breath Hold	Peak inspiratory breath hold. Adjust APL valve to maintain circuit pressure at desired PEEP for 5–10 s before beginning advanced imaging sweep.
Biopsy	Ensure ventilator settings are the same as those when performing sweep.

APL: adjustable pressure-limiting valve, FiO_2_: fraction of inspired oxygen, PEEP: positive end expiratory pressure, TIVA: total intravenous anesthesia.

## Data Availability

Not applicable.

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
