# Peer review of "Advanced Imaging for Robotic Bronchoscopy: A Review"

_diagnostics, 2023, doi:10.3390/diagnostics13050990_

Round 1
Reviewer 1 Report
The paper is well written and easy flow of thoughts. It is also concise and informative. I recommend publishing this paper.
One minor thing to please add:
Please add the ablation modality used in line 340-342.
Reviewer 2 Report
An excellent overview of the techniques and capabilities of robotic bronchoscopy. A very carefully prepared manuscript. It is full of specific technical information that is crucial in this particular case. It is also great that the information is presented objectively, impartially, and critically. Congratulations to the authors for their high-quality work. I have no critical comments.
Reviewer 3 Report
Ravikumar et al wrote a review on the recent advances in adjunct imaging platforms and protocols with navigational bronchoscopy.
1. Please kindly see if a clearer figure 5 can be included in the manuscript.
2. For ease of reading, please include the actual diagnostic yield improvement (if available) for the statement linked to Reference 26.
3. It might be beneficial to the non-expert readers to briefly introduce the differences between Ion & Monarch RAB platform before going into Section 3.2 & 3.3.
Reviewer 4 Report
The authors provide a detailed and extensive review of the current status of various devices in the diagnosis of lung tumours.
The descriptions are excellent and I consider it suitable for acceptance in the Journal of Diagnostics.
